# High prevalence of burnout syndrome among medical and nonmedical residents during the COVID-19 pandemic

**Rebeca da Nóbrega Lucena Pinho**[1]*, **Thais Ferreira Costa**[2], **Nayane Miranda Silva**[1], **Adriana Ferreira Barros-Areal**[3], **André de Matos Salles**[4], **Andrea Pedrosa Ribeiro Alves Oliveira**[5], **Carlos Henrique Reis Esselin Rassi**[6], **Ciro Martins Gomes**[7], **Dayde Lane Mendonça da Silva**[8], **Fernando Araújo Rodrigues de Oliveira**[9], **Isadora Jochims**[10], **Ivan Henrique Ranulfo Vaz Filho**[11], **Lucas Alves de Brito Oliveira**[12], **Marta Alves Rosal**[13], **Marta Pinheiro Lima**[14], **Mayra Veloso Ayrimoraes Soares**[10], **Patricia Shu Kurizky**[15], **Viviane Cristina Uliana Peterle**[16], **Ana Paula Monteiro Gomides**[17], **Licia Maria Henrique da Mota**[18], **Cleandro Pires de Albuquerque**[19], **Cezar Kozak Simaan**[20], **Veronica Moreira Amado**[5]

1 Hospital Universitário de Brasília—HUB-UnB, Universidade de Brasília–UnB, Brasília–DF, Brazil, 2 Secretaria de Estado de Saúde do Distrito Federal–SES DF, Brasília–DF, Brazil, 3 Neurologista, Doutoranda pós-graduação em ciências médicas-UnB, Preceptora de graduação Escola Superior de Ciências da Saúde- ESCS, Brasília–DF, Brazil, 4 Psiquiatra da Infância e Adolescência do Hospital Universitário de Brasília—HUB/UnB, Brasília–DF, Brazil, 5 Professor of Medical Faculty, University of Brasília–UnB, Brasília–DF, Brazil, 6 Hospital Universitário de Brasília da Universidade de Brasília (HUB-UnB) e Hospital Sírio-Libanês, Brasília–DF, Brazil, 7 Programa de Pós-Graduação em Ciências Médicas e Núcleo de Medicina Tropical, Faculdade de Medicina, Universidade de Brasília, Brasília–DF, Brazil, 8 Professora Adjunta do Departamento de Farmácia—UnB e gerente de ensino e pesquisa HUB-UnB, Brasília–DF, Brazil, 9 Universidade de Brasília–UnB, Brasília–DF, Brazil, 10 Hospital Universitário de Brasília da Universidade de Brasília—HUB-UnB, Brasília–DF, Brazil, 11 Doutorando Programa de Pós-Graduação em Ciências Médicas FM-UnB, Brasília–DF, Brazil, 12 Faculdade de Medicina, Universidade de Brasília FM-UnB, Brasília–DF, Brazil, 13 Professora Adjunta da Disciplina de Ginecologia da UFPI; Coordenadora da COREME do HU-UFPI, Teresina–PI, Brazil, 14 Empresa Brasileira de Serviços Hospitalares–EBSERH, Brasília–DF, Brazil, 15 Serviço de Dermatologia do Hospital Universitário de Brasília e Programa de pós-graduação em ciências médicas da UnB, Brasília–DF, Brazil, 16 Escola Superior de Ciências da Saúde/ESCS, Brasília–DF, Brazil, 17 Centro Universitário de Brasília–Uniceub, Brasília–DF, Brazil, 18 Docente do programa de pós-graduação em Ciências Médicas da Faculdade de Medicina da Universidade de Brasília, Médica Reumatologista do Hospital Universitário de Brasília—HUB-UNB-EBSERH, Brasília–DF, Brazil, 19 Hospital Universitário de Brasília—UnB, Programa de Pós-graduação em Ciências Médicas/FM-UnB, Brasília–DF, Brazil, 20 Professor da Disciplina de Reumatologia da UnB, Brasília–DF, Brazil

* nlp.rebeca@gmail.com

## Abstract

### Background

Since the beginning of the COVID-19 pandemic, health professionals have been working under extreme conditions, increasing the risk of physical and mental illness. We evaluated the prevalence of burnout and its associated factors among postgraduate student residents in health professions during the global health crisis.

### Methods

Healthcare residents were recruited from all across Brazil between July and September 2020 through digital forms containing instruments for assessing burnout (Oldenburg

**Data Availability Statement:** All relevant data are within the paper and its Supporting Information files.

**Funding:** This study was funded in part by the University of Brasília (UnB). No additional external funding was received for this study.

**Competing interests:** The authors have declared that no competing interests exist.

Burnout Inventory (OLBI)), resilience (brief resilient coping scale (BRCS)) and anxiety, stress and depression (depression, anxiety and stress scale (DASS-21) and Patient Health Questionnaire (PHQ-9)). Additionally, the relationships between burnout and chronic diseases, autonomy and educational adequacy in the residency programme, personal protective equipment (PPE), workload and care for patients with COVID-19 were evaluated. The chi-square test, Student's t test, Pearson's correlation test and logistic regression were performed.

## Results

A total of 1,313 participants were included: mean (standard deviation) age, 27.8 (4.4) years; female gender, 78.1%; white race, 59.3%; and physicians, 51.3%. The overall prevalence of burnout was 33.4%. The odds (odds ratio [95% confidence interval]) of burnout were higher in the presence of pre-existing diseases (1.76 [1.26–2.47]) and weekly work > 60 h (1.36 [1.03–1.79]) and were lower in the presence of high resilience (0.84 [0.81–0.88]), autonomy (0.87 [0.81–0.93]), and educational structure (0.77 [0.73–0.82]), adequate availability of PPE (0.72 [0.63–0.83]) and non-white race (0.63 [0.47–0.83]). Burnout was correlated with anxiety (r = 0.47; p < 0.05), stress (r: 0.58; p < 0.05) and depression (r: 0.65; p < 0.05).

## Conclusions

We observed a high prevalence of burnout among residents during the COVID-19 pandemic. Individual characteristics and conditions related to the work environment were associated with a higher or lower occurrence of the syndrome.

## Introduction

In early 2020, COVID-19, caused by a new coronavirus (SARS-CoV-2), spread rapidly throughout the world and reached pandemic status, requiring the rapid and extensive reorganization of health services [1–3]. There were many challenges to which health professionals were imposed, such as uncertainties regarding the magnitude, duration and global effects of the health crisis, the level of preparation of individuals and of the institutions to cope with the crisis, and the risk of infection, which could be life threatening. In this context of insecurity, anxiety and work overload, the risk of physical and mental illness among this population is a concern [4].

Brazil was the first South American country to report a confirmed case of COVID-19 (02/26/2020) [5]. The epidemiological scenario soon became dramatic, with uncontrolled growth in the number of confirmed cases and deaths, tending to the collapse of national health systems [6]. On 10/07/2021, the country surpassed 600,000 deaths due to the disease, becoming the nation with the second highest number of deaths in the world, behind only the United States of America [7].

Medical and nonmedical health residencies involve extensive programme content and a high weekly workload. In the context of the pandemic, the prolonged and uncomfortable use of personal protective equipment (PPE), irregular hydration and feeding and sleep deprivation increased fatigue and the risk of burnout [3]. Burnout is defined as a multifaceted construct characterized by emotional exhaustion, depersonalization and a low sense of personal accomplishment [8].

The literature on this topic is still scarce despite the importance of understanding the impact of the pandemic on health professionals in training and estimating the prevalence of

burnout and its relationship with other mental conditions such as stress, depression and anxiety, thus contributing to the development of alternatives that mitigate this problem.

The objective of the present study was to evaluate the prevalence of burnout syndrome among health professionals in training, medical residencies and other health areas and to identify the factors associated with the occurrence of burnout in this specific population.

## Materials and methods

This study served as the baseline evaluation of a longitudinal study still in progress, whose protocol has been published; the study included post-graduate student residents in health professions, aged over 18 years, assigned to the direct provision of care to patients during the COVID-19 pandemic and sought to identify risk factors associated with burnout in this population [9].

Recruitment occurred via e-mail, messages on social networks, posters in hospitals and the university hospital intranet containing *QR codes* with *links* to the survey forms. The codes and links were sent to the 7,215 residents of 40 university hospitals affiliated with the Brazilian Hospital Services Company (Empresa Brasileira de Serviços Hospitalares—EBSERH). EBSERH is a public company linked to the Ministry of Education established to manage federal university hospitals. Health professional residents at any healthcare institution in the country could also participate.

Data collection was performed using a structured electronic form (via Microsoft Forms) designed to gather information on the clinical and epidemiological characteristics of the participants; the form also included the assessment instruments used in the study, in accordance with the approved protocol [9]. The following instruments were applied.

**1. Oldenburg Burnout Inventory (OLBI):** This instrument has been adapted for and validated in Portuguese for the evaluation of burnout and contains eight questions in each of the "disengagement" (OLBI-D) and "exhaustion" (OLBI-E) subscales, totalling 16 questions (OLBI Total); responses are provided using a five-point Likert scale. Disengagement refers to distancing from work and the development of work-related cynical and negative attitudes and behaviours. Exhaustion refers to feelings of physical fatigue, need for rest, and feelings of overload and work-related emptiness [10]. We adopted the method used by Delgadillo et al. [11], who defined the cut-off point (values equal to or greater than the mean + 1 standard deviation) for the classification of the total OLBI score as "high", thus indicative of burnout [11]. We applied this method using values observed in the Brazilian population [10].

**2. Brief resilient coping scale (BRCS):** This is a one-dimensional instrument adapted for and validated in Portuguese consisting of four items that assess the ability to adaptively cope with stress [12]. In this study, a score less than 13 was considered "low resilience".

**3. Degree of autonomy to decide behaviours at work:** A visual numerical scale was used to evaluate each individual's perception of his or her degree of autonomy at work. The responses ranged from 0 to 10 (0 "I have no autonomy" and 10 "I have full autonomy"). A value ≤ 4 indicated a low perception of autonomy at work.

**4. Adequacy of the educational organization of the residency programme:** A visual numerical scale was used to evaluate each individual's perception of the adequacy of the educational structure of his or her residency programme. The responses ranged from 0 to 10, with 0 being "completely inadequate" and 10 being "completely adequate". The cut-off point defined for "poor educational adequacy" was ≤ 5.

**5. Availability of PPE:** A 5-point Likert scale was used to evaluate the perception of residents regarding the adequacy of PPE availability in their professional practice. The following question was asked: "In your professional practice, in patient care, how often do you have sufficient and adequate PPE available?". The possible responses were as follows: 1—at no time, 2

—less than half the time, 3—half the time, 4—more than half the time and 5—all the time. The cut-off point for "inadequate PPE availability" was defined as $\leq 3$.

   **6. External work link:** Respondents provided an answer of YES or NO regarding the exercise of professional work outside of the residency programme.

   **7. Providing direct care to patients with COVID-19:** Respondents provided an answer of YES or NO as to whether, in their practice in the residency programme, direct care was mandatory for patients with COVID-19.

   **8. Depression, anxiety and stress scale (DASS-21):** This instrument has been translated into and validated for Portuguese [13] and is composed of three subscales covering the domains of depression (DASS21-D), anxiety (DASS21-A) and stress (DASS21-S), with cut-off points $> 9$, $> 7$ and $> 14$, respectively, for the classification of scores as "high", thus indicative of the respective mental disorders.

   **9. Brief Depression Scale (Patient Health Questionnaire/PHQ-9):** This instrument has been translated into and validated for Brazil [14]. It consists of nine questions that assess the frequency of depressive symptoms. The cut-off score for the classification of the scores as high, thus indicative of depressive disorder, was defined as $\geq 9$.

   The sample size was calculated considering the objectives of the longitudinal study, which is still in progress [9] and seeks to establish the incidence of burnout and identify its predictors among residents during the COVID-19 pandemic, corresponding to a cross-sectional evaluation of data obtained at the baseline of a longitudinal follow-up study. The sample size was calculated based on the following parameters:

1. an expected prevalence of burnout of 28% among health professionals [15];

2. an expected difference of 10 percentage points in the incidence of burnout between the exposure and control groups after 12 weeks of follow-up; and

3. the offset of losses to follow-up (approximately 20%).

   Thus, the minimum sample size was calculated as N = 1144 participants.

   The data for the sample are provided as absolute and relative frequencies for categorical variables and as measures of central tendency and dispersion for continuous numerical variables. In bivariate analyses, associations between dichotomous categorical variables were verified using the chi-square test, with odds ratios and Cramer's V used to estimate effect sizes. Differences between groups regarding continuous variables were verified by Student's t test, with Welch correction for nonhomogeneous variances. Correlations were verified using Pearson's r coefficient. Binomial logistic regression models were used to identify the presence of burnout and evaluate the independent contribution of several candidate predictor variables. Predictor variables that were significant in the bivariate analyses were incorporated into the multivariate analysis. Values of $p < 0.05$ were considered significant. The analyses were conducted in SPSS 25.

   The study was approved by a local research ethics committee and the National Research Ethics Committee (Comitê de Ética em Pesquisa/Comissão Nacional de Ética em Pesquisa–CEP/CONEP), available at https://plataformabrasil.saude.gov.br/, under registration number CAAE: 33493920.0.0000.5558. All participants signed and received a copy of the informed consent form via e-mail.

## Results

A total of 1,313 residents responded to the survey. The respondents were residents at 135 public, private and philanthropic health institutions from 25 federal units; 89.6% were affiliated with university hospitals.

The sample consisted of medical residents (51.3%, n = 674), nurses (8.8%, n = 115), pharmacists (6.9%, n = 91), nutritionists (6.2%, n = 82), psychologists (6.2%, n = 82), physical therapists (4.8%, n = 63), social workers (3.9%, n = 51), dentists (2.8%, n = 37), occupational therapists (1.7%, n = 22), and other residents (4.2%, n = 55). Among the participants, there was a predominance of the female gender and white race (Table 1). The provision of direct care to patients with COVID-19 was reported by 60.2% of the residents. Of the total, 17.8%

**Table 1. General characteristics of the study population.**

| Characteristics | Total |
|---|---|
| | **n = 1313** |
| **Gender** | |
| Female | 1025 (78.2%) |
| Male | 285 (21.8%) |
| **Race** | |
| White | 778 (59.3%) |
| Non-white | 535 (40.7%) |
| **Nature of the educational institution** | |
| Public | 1277 (97.2%) |
| Private or philanthropic | 36 (2.8%) |
| **University hospital** | |
| Yes | 1177 (89.6%) |
| No | 136 (10.4%) |
| **Category of professional participant (1272 responses)** | |
| Physician | 674 (53.0%) |
| Other health professional | 598 (47%) |
| **Providing direct care to patients with COVID-19** | |
| Yes | 790 (60.2%) |
| No | 523 (39.8%) |
| **Presence of diseases (n = 1305)** | |
| Yes | 234 (17.9%) |
| No | 1071 (82.1%) |
| **Increased risk for severe forms of COVID-19** | |
| Yes | 218 (16.7%) |
| No | 1087 (83.3%) |
| **Perception of PPE availability** | |
| Poor availability | 281 (21.4%) |
| Moderate to good availability | 1032 (78.6%) |
| **Perception of the educational organization of the residency programme** | |
| Poor adequacy | 558 (42.5%) |
| Moderate to good adequacy | 755 (57.5%) |
| **Autonomy to decide behaviours at work** | |
| Low autonomy | 224 (17.1%) |
| Moderate to high autonomy | 1089 (82.9%) |
| **Weekly workload** | |
| $\leq$ 60 h | 541 (41.2%) |
| $\geq$ 60 h | 772 (58.8%) |
| **Activity outside the residency programme** | |
| Yes | 424 (32.3%) |
| No | 889 (67.7%) |

reported having pre-existing diseases, among whom 93.1% were classified as being at increased risk for severe forms of COVID-19 [16].

Regarding the weekly workday, 58.8% worked ≥ 60 hours per week; 67.7% did not work outside the residency programme; 78.6% reported that the adequacy of the availability of PPE for the provision of health care was moderate to good perception; 42.5% reported that the adequacy of the educational organization of their residency programme was poor; and 17.1% indicated low autonomy in deciding work behaviours (Table 1).

The mean (SD) age, for the overall sample, was 27.8 (4.4) years, and the mean scores for the instruments were as follows: OLBI-D, 2.8 (0.8); OLBI-E, 3.6 (0.7); OLBI Total, 3.2 (0.7); BRCS, 12.4 (3.8); DASS-21 depression, 15.3 (11.3); DASS-21 anxiety, 12.1 (10.3); DASS-21 stress, 20.3 (10.7); PHQ-9, 12.0 (6.5); perception of autonomy, 6.5 (2.1); and adequacy of the educational structure, 5.8 (2.5).

A moderate to strong positive correlation was observed between DASS-21 anxiety and OLBI-E (r: 0.48 and $p < 0.05$) and OLBI-Total (r: 0.47 and $p < 0.05$); between DASS-21 stress and OLBI-D (r: 0.46 and $p < 0.05$), OLBI-E (r: 0.57 and $p < 0.05$) and OLBI-Total (r: 0.58 and $p < 0.05$); and between PHQ-9 (depression) and OLBI-D (r: 0.53 and $p < 0.05$), OLBI-E (r: 0.64 and $p < 0.05$) and OLBI-Total (r: 0.65 and $p < 0.05$).

Table 2 shows the differences between medical residents and nonmedical residents regarding the scores obtained for the instruments used to evaluate resilience (BRCS), distancing (OLBI-D), exhaustion (OLBI-E), burnout (OLBI-Total), perception of autonomy and adequacy of the educational structure.

The mean age of the medical residents was 29.2 (4.4) years, and that of the nonmedical residents was 26.4 (4.0) years ($p < 0.001$). Low resilience was found in more than half of the participants in both types of residency programmes (Table 3). Medical residents considered the educational structure of their residency programme more adequate than did nonmedical residents. The number of medical residents who had work activity outside the training programme and who provided direct care to patients with COVID-19 was significantly higher than that of nonmedical residents (Table 3).

The overall prevalence of burnout in our study was 33.4%. There was a significant association between burnout and the variables race, presence of pre-existing diseases, perception of autonomy, perception of adequacy of the educational structure, perception of availability of PPE, weekly workload and low resilience (Table 4).

There was no difference in the degree of resilience (BRCS) between genders (low resilience: female 62.3% [n = 639], male 60.4% [n = 172]; p = 0.540; OR 1.08; 95% CI 0.83–1.42) or between races (white 61.6% [n = 479], non-white 62.4% [n = 334]; p = 0.752; OR 1.03; 95% CI 0.83–1.3).

**Table 2. Scores for medical and nonmedical residents on the instruments used to assess resilience, distancing, exhaustion, burnout, perception of autonomy and adequacy of the educational structure.**

| Variable | Nonmedical* (n = 639) | Medical* (n = 674) | Difference in means | [95% CI] | p** |
|---|---|---|---|---|---|
| BRCS Resilience | 12.26 (3.65) | 12.56 (3.88) | -0.30 | [-0.71; 0.11] | 0.155 |
| OLBI Distancing | 2.74 (0.81) | 2.81 (0.86) | -0.07 | [-0.16; 0.02] | 0.115 |
| OLBI Exhaustion | 3.58 (0.69) | 3.53 (0.78) | 0.05 | [-0.03; 0.13] | 0.208 |
| OLBI Total | 3.16 (0.66) | 3.17 (0.74) | - 0.01 | [-0.09; 0.06] | 0.771 |
| Perception of autonomy | 6.58 (2.10) | 6.43 (2.12) | 0.15 | [-0.08; 0.38] | 0.191 |
| Adequacy of the educational structure | 5.34 (2.50) | 6.18 (2.37) | -0.84 | [-1.10; -0.58] | < 0.001 |

* The values in the table are the mean (standard deviation).

**Significance level (p value) based on Student's t test.

**Table 3. Differences between medical residents and nonmedical residents regarding the various characteristics studied—bivariate analyses (unadjusted).**

| Variable or outcome | Nonmedical n (%) | Medical n (%) | Odds ratio [95% CI] | p* |
|---|---|---|---|---|
| **Gender (n = 1310)** | | | | |
| Female | 549 (86.3%) | 476 (70.6%) | 2.63 [1.98–3.47] | < 0.001 |
| **Race (n = 1313)** | | | | |
| White | 338 (52.9%) | 440 (65.3%) | 0.60 [0.48–0.75] | < 0.001 |
| **BRCS—Resilience (n = 1313)** | | | | |
| Low | 414 (64.8%) | 399 (59.2%) | 1.26 [1.01–1.58] | 0.037 |
| **OLBI–Burnout (n = 1313)** | | | | |
| High | 202 (31.6%) | 236 (35%) | 0.86 [0.68–1.08] | 0.191 |
| **Autonomy (n = 1313)** | | | | |
| Moderate/High | 535 (83.7%) | 554 (82.2%) | 1.11 [0.84–1.49] | 0.462 |
| **Educational structure (n = 1313)** | | | | |
| Adequate | 312 (48.8%) | 443 (65.7%) | 0.50 [0.40–0.62] | < 0.001 |
| **Availability of PPE (n = 1313)** | | | | |
| Moderate/High | 515 (80.6%) | 517 (76.7%) | 1.26 [0.97–1.64] | 0.086 |
| **Weekly workload (n = 1313)** | | | | |
| >60 h | 294 (46%) | 478 (70.9%) | 0.35 [0.28–0.44] | < 0.001 |
| **Activity outside the residency programme (n = 1313)** | | | | |
| Yes | 8 (1.3%) | 416 (61.7%) | 0.01 [0.00–0.02] | < 0.001 |
| **Direct care for patients with COVID-19 (n = 1313)** | | | | |
| Yes | 246 (38.5%) | 544 (80.7%) | 0.15 [0.12–0.19] | < 0.001 |

* Significance level (p value) based on the chi-square test.

There were differences between genders regarding the type of residency programme, weekly workload, activity outside the residency programme and direct provision of care to patients with COVID-19 (Table 5).

All individual characteristics that were significantly associated with burnout in the unadjusted (bivariate) analyses remained significant independent predictors of burnout syndrome in the multivariate analysis by logistic regression (Fig 1).

## Discussion

The aim of this study was to evaluate the prevalence of burnout in post-graduate student residents in health professions in Brazil in the context of the COVID-19 pandemic. A notable aspect of this study is that it comparatively evaluates different residency programmes, reinforcing that data on nonmedical health residencies are scarce in the scientific literature.

Our sample had similar representativeness regarding the number of medical residents (n = 674) and nonmedical residents (n = 639). In a survey conducted on the website of the Ministry of Education (Committees of Medical and Multiprofessional Residencies in Health), no data were available on the composition of health residency programmes in Brazil regarding gender.

In health programmes, women represent the majority gender in post-graduate programmes in general [17]. However, there is a lack of data in studies with a methodology similar to that used herein. Almeida et al. [18] stated that females are more vulnerable to mental health problems, such as higher levels of stress, anxiety, depression and posttraumatic stress symptoms. Furthermore, there are data that indicate that women seek health services twice as often as men [19], which may justify a greater interest in participating in scientific research focused on mental health and the prevention of future problems.

**Table 4. Association between burnout (OLBI) and various characteristics of the participants—bivariate analyses (not adjusted).**

| Variable | Burnout frequencies* | Odds ratio | p ** |
|---|---|---|---|
| | n (%) | [95% CI] | |
| **Gender (n = 1310)** | | | |
| Male | 94 (33%) | 1.02 | 0.903 |
| Female | 342 (33.4%) | [0.77–1.35] | |
| **Race (n = 1313)** | | | |
| White | 279 (35.9%) | 0.76 | 0.020 |
| Non -white | 159 (29%.7) | [0.60–0.96] | |
| **Presence of diseases (n = 1305)** | | | |
| No | 331 (30.9%) | 1.82 | <0.001 |
| Yes | 105 (44.9%) | [1.36–2.43] | |
| **Autonomy to decide behaviours at work (n = 1313)** | | | |
| Low | 132 (58.9%) | 0.27 | <0.001 |
| Moderate/High | 306 (28.1%) | [0.20–0.37] | |
| **Perception of the educational organization of the residency programme (n = 1313)** | | | |
| Inadequate | 280 (50.2%) | 0.26 | <0.001 |
| Adequate | 158 (20.9%) | [0.21–0.34] | |
| **Perception of adequacy of PPE availability (n = 1313)** | | | |
| Low | 140 (49.8%) | 0.41 | <0.001 |
| Moderate/High | 298 (28.9%) | [0.31–0.54] | |
| **Weekly workload (n = 1313)** | | | |
| ≤ 60 h | 161 (29.8%) | 1.32 | 0.021 |
| > 60 h | 277 (35.9%) | [1.04–1.67] | |
| **Activity outside the residency programme (n = 1313)** | | | |
| No | 291 (32.7%) | 1.09 | 0.487 |
| Yes | 147 (34.7%) | [0.85–1.39] | |
| **Direct provision of care to patients with COVID-19 (n = 1313)** | | | |
| No | 171 (32.7%) | 1.05 | 0.679 |
| Yes | 267 (33.8%) | [0.83–1.33] | |
| **BRCS—Resilience (n = 1313)** | | | |
| Moderate/High | 93 (18.6%) | 3.23 | < 0.001 |
| Low | 345 (42.4%) | [2.48–4.20] | |

* OLBI score ≥ mean + 1 SD unit

** Significance level (p value) according to the chi-square test.

Among health professionals working at the forefront of epidemic care, being female, of a younger age [20, 21] and in training [22] are risk factors for mental disorders, especially burnout. However, in our study, there was no association between burnout prevalence and gender or the provision of direct care to patients with COVID-19. Mental disorders, in general, are more frequent among women; biological, cultural and social components, such as overload resulting from double work shifts (family and external) and high socio-family demands, are indicated as predisposing factors for the emergence of psychological disorders in the female population [23].

The overall prevalence of burnout in our sample was 33.3%, with no significant differences between medical residents and nonmedical residents (35% vs. 31.6%, p = 0.191). Da Cruz Gouveia et al. [24] reported a prevalence of 27.9% in a Brazilian study that described the factors

**Table 5. Differences between genders regarding the various variables studied.**

| Variable | Male | Female | Odds ratio | p* |
|---|---|---|---|---|
| | n (%) | n (%) | [95% CI] | |
| **RACE (n = 1310)** | | | | |
| Non-white | 124 (43.5%) | 410 (40.0%) | 0.87 | 0.286 |
| | | | [0.66–1.13] | |
| **Presence of illness (n = 1302)** | | | | |
| Yes | 57 (20.2%) | 175 (17.2%) | 0.82 | 0.235 |
| | | | [0.59–1.14] | |
| **Type of residency programme (n = 1310)** | | | | |
| Physician | 198 (69.5%) | 476 (46.4%) | 0.38 | < 0.001 |
| | | | [0.29–0.50] | |
| **BRCS—Resilience (n = 1310)** | | | | |
| Low | 172 (60.4%) | 639 (62.3%) | 1.09 | 0.540 |
| | | | [0.83–1.42] | |
| **Autonomy to decide behaviours at work (n = 1310)** | | | | |
| Moderate/High | 226 (79.3%) | 861 (84.0%) | 1.37 | 0.062 |
| | | | [0.98–1.91] | |
| **Perception of adequacy of the educational structure (n = 1310)** | | | | |
| Adequate | 162 (56.8%) | 592 (57.8%) | 1.04 | 0.782 |
| | | | [0.80–1.35] | |
| **Perception of adequacy of PPE availability (n = 1310)** | | | | |
| Moderate/High | 219 (76.8%) | 811 (79.1%) | 1.14 | 0.406 |
| | | | [0.84–1.56] | |
| **Weekly workload (n = 1310)** | | | | |
| >60 h | 185 (64.9%) | 585 (57.1%) | 0.72 | 0.017 |
| | | | [0.55–0.94] | |
| **Activity outside the residency programme (n = 1310)** | | | | |
| Yes | 142 (49.8%) | 281 (27.4%) | 0.38 | < 0.001 |
| | | | [0.29–0.50] | |
| **Direct provision of care to patients with COVID (n = 1310)** | | | | |
| Yes | 213 (74.7%) | 575 (56.1%) | 0.43 | < 0.001 |
| | | | [0.32–0.58] | |

* Significance level (p value) according to the chi-square test

associated with burnout syndrome in residents of a university hospital in a pre-pandemic period [24]. In a recent study conducted in Japan to evaluate the prevalence of burnout in health professionals during the COVID-19 pandemic, an overall prevalence of 31.4% was reported [25].

In our study, the individual characteristics independently associated (multivariate analysis) with a higher prevalence of burnout were the presence of chronic diseases and weekly workload > 60 h, and those associated with a lower prevalence of burnout were non-whites, perception of greater autonomy to decide behaviours at work, perception of an adequate educational structure of the residency programme, adequate availability of PPE, and greater resilience.

The emergence of burnout results from work overload and often occurs during the first two years of resident training, occurring cumulatively in up to 74% of resident physicians [26]. The results from a Chinese study conducted with medical professionals and nurses to assess

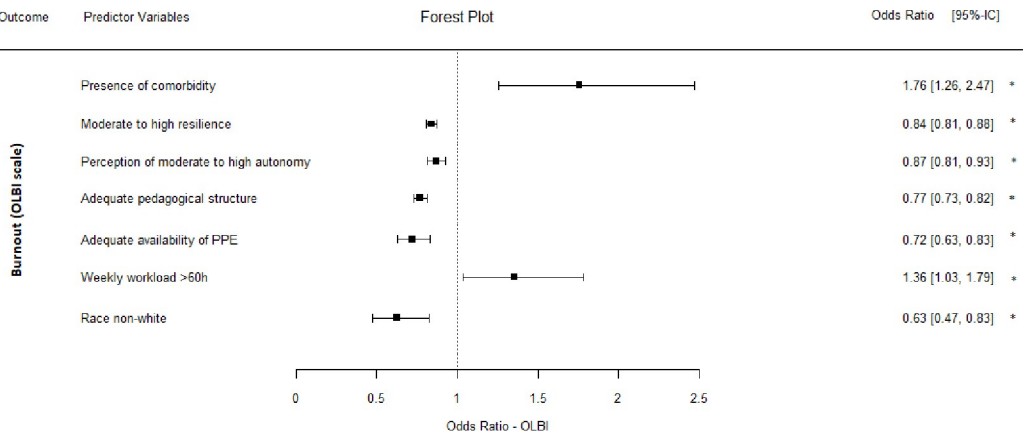

**Fig 1. Multivariate logistic regression analysis of predictors associated with burnout (OLBI) among health residents.**

burnout outside of a pandemic context indicate that long working hours contribute to the occurrence of burnout [27].

There is evidence in the scientific literature of a reduction in the prevalence of burnout among residents after the implementation of limits to working hours. In a study with 118 residents and interns, those who worked > 80 hours per week had a burnout prevalence 31% higher than that for those who worked < 80 hours per week [28]. Although burnout is usually attributed to high demands or work-related stress, the impact of excessively high workloads cannot be neglected. In our study, more than half of the participants (51.9%) had a weekly workday that exceeded 60 hours, which was a predictor of burnout. Significant differences were found between the types of residency programmes with respect to the variables activity outside the residency programme and weekly workload. This difference is justified because nonmedical residents must dedicate themselves exclusively to the residency programme and cannot perform work activities outside the programme [29].

In our study, we did not observe an association between burnout and the direct provision of care to patients with COVID-19, despite greater exposure to the risk of infection resulting from this activity. A study conducted in Iran with 266 nurses evaluated the level of burnout during the COVID-19 pandemic; in front line and non-front line workers, the work stress and burnout scores for the group exposed to COVID-19 were significantly higher than those for the non-exposure group [30]. The differences between the populations studied and the working conditions between the two countries may contribute to explaining the differences observed in the studies.

Additionally, in our study, the presence of pre-existing diseases in residents increased the chance of developing burnout by 76%. Consistent with this finding, Lo et al. [31] suggested that health problems such as viral and respiratory infections, diabetes, cardiovascular diseases, obesity and liver diseases can result from burnout. This study was conducted with workers from a monitor manufacturing company in central Taiwan [31].

A strong correlation was observed between the presence of burnout and the poor adequacy of the educational structure of the programmes, which was reported in high percentages by both medical and nonmedical participants (34.3% and 51.2%, respectively). This finding reinforces the fact that burnout is also driven by organizational factors in addition to individual factors. Other factors related to training, such as the high demand for learning in relatively

short periods of time and the strict supervision of behaviour (limitation of autonomy), represent additional risks for residents compared to physicians [32].

The low perception of PPE availability also had an impact on the development of burnout in our study. Consistent with this finding, a study conducted to identify factors that contribute to burnout among health professionals during the COVID-19 pandemic found that available and adequate PPE was considered a protective factor for burnout and that a lack of PPE was a causative agent of stress [1].

The mean BRCS score was numerically lower (suggesting a lower degree of resilience) among nonmedical residents than among medical residents, although the difference did not reach statistical significance (Table 2). However, when evaluating the proportions of individuals with low resilience, there was a significant difference between the groups, with a higher frequency of low resilience among non-physicians (Table 3).

Regarding the limitations of the study, we recognize the possibility of selection bias towards individuals who agreed to participate in the study. The findings do not necessarily reflect the reality of individuals who chose not to participate. However, participation is voluntary in any clinical study. Thus, the possibility of not reflecting those who chose not to participate is inherent to any survey and not only to this study.

In addition, there was a clear predominance of responses from residents associated with university hospitals, which generally have a better educational and physical structure than do most non-profit, non-university hospitals of similar size (with some exceptions). Therefore, the reality of non-university hospitals may not be adequately reflected in the data from this study.

The study is also limited by the exclusive use of digital forms for remote data collection and the use of validated instruments for the evaluation of burnout, resilience, anxiety, stress and depression, without in-person clinical evaluations for the confirmation of the diagnoses suggested by the instruments.

Despite the limitations mentioned, the results of this study may be useful for developing strategies to prevent or mitigate the damage caused by burnout among residents and provide better working conditions and support for the mental health of these professionals in training.

## Conclusions

The results of this study indicate a high prevalence of burnout among health professionals in training in the context of the COVID-19 pandemic. Individual characteristics as well as those related to working conditions are associated with the occurrence of burnout in this population.

## Supporting information

**S1 Data.**
(XLSX)

**S1 File.**
(PDF)

## Acknowledgments

We would like to thank the University Hospital of Brasília, especially the Superintendency and the Division of Teaching and Research, and EBSERH for the support provided to this study.

## Author Contributions

**Conceptualization:** Rebeca da Nóbrega Lucena Pinho, Nayane Miranda Silva, Licia Maria Henrique da Mota, Cleandro Pires de Albuquerque, Cezar Kozak Simaan, Veronica Moreira Amado.

**Data curation:** Cleandro Pires de Albuquerque.

**Formal analysis:** Rebeca da Nóbrega Lucena Pinho, Nayane Miranda Silva, Licia Maria Henrique da Mota, Cleandro Pires de Albuquerque, Cezar Kozak Simaan, Veronica Moreira Amado.

**Investigation:** Dayde Lane Mendonça da Silva, Fernando Araújo Rodrigues de Oliveira, Marta Alves Rosal, Viviane Cristina Uliana Peterle.

**Methodology:** Rebeca da Nóbrega Lucena Pinho, Nayane Miranda Silva, Licia Maria Henrique da Mota, Cleandro Pires de Albuquerque, Cezar Kozak Simaan, Veronica Moreira Amado.

**Project administration:** Dayde Lane Mendonça da Silva, Fernando Araújo Rodrigues de Oliveira, Licia Maria Henrique da Mota, Veronica Moreira Amado.

**Resources:** Rebeca da Nóbrega Lucena Pinho, Thais Ferreira Costa, Nayane Miranda Silva, Adriana Ferreira Barros-Areal, André de Matos Salles, Andrea Pedrosa Ribeiro Alves Oliveira, Carlos Henrique Reis Esselin Rassi, Ciro Martins Gomes, Dayde Lane Mendonça da Silva, Fernando Araújo Rodrigues de Oliveira, Isadora Jochims, Ivan Henrique Ranulfo Vaz Filho, Lucas Alves de Brito Oliveira, Marta Alves Rosal, Marta Pinheiro Lima, Mayra Veloso Ayrimoraes Soares, Patricia Shu Kurizky, Viviane Cristina Uliana Peterle, Ana Paula Monteiro Gomides, Licia Maria Henrique da Mota, Cleandro Pires de Albuquerque, Cezar Kozak Simaan, Veronica Moreira Amado.

**Software:** Rebeca da Nóbrega Lucena Pinho, Nayane Miranda Silva, Cleandro Pires de Albuquerque.

**Supervision:** Licia Maria Henrique da Mota, Veronica Moreira Amado.

**Validation:** Adriana Ferreira Barros-Areal, André de Matos Salles, Andrea Pedrosa Ribeiro Alves Oliveira, Ciro Martins Gomes, Patricia Shu Kurizky, Ana Paula Monteiro Gomides.

**Visualization:** Thais Ferreira Costa, Carlos Henrique Reis Esselin Rassi, Isadora Jochims.

**Writing – original draft:** Rebeca da Nóbrega Lucena Pinho.

**Writing – review & editing:** Thais Ferreira Costa, Adriana Ferreira Barros-Areal, André de Matos Salles, Ivan Henrique Ranulfo Vaz Filho, Lucas Alves de Brito Oliveira, Marta Pinheiro Lima, Mayra Veloso Ayrimoraes Soares, Licia Maria Henrique da Mota, Cleandro Pires de Albuquerque, Cezar Kozak Simaan, Veronica Moreira Amado.

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
