## [Decision Letter · Decision Letter 0]

19 Jul 2022

PONE-D-22-10606High prevalence of burnout syndrome among medical and nonmedical residents during the COVID-19 pandemicPLOS ONE

Dear Dr. Pinho,

Thank you for submitting your manuscript to PLOS ONE. After careful consideration, we feel that it has merit but does not fully meet PLOS ONE’s publication criteria as it currently stands. Therefore, we invite you to submit a revised version of the manuscript that addresses the points raised during the review process.

We look forward to receiving your revised manuscript.

Kind regards,

Mohammad Hossein Ebrahimi

Academic Editor

PLOS ONE

Journal Requirements:

We would like to thank the University Hospital of Brasília, especially the Superintendency and the Division of Teaching and Research, and EBSERH for the support provided to this study.

4. Please upload a copy of Figure 1, to which you refer in your text on page xx. If the figure is no longer to be included as part of the submission please remove all reference to it within the text.

Additional Editor Comments:

1. Review Comments to the Author

Reviewer #1: This study addresses an important issue concerning the COVID-19 impact. Nevertheless, I do have several concerns in this particular study I would like to address.

Comments in the article which needs clarification, rewrites and/or additional information

Title:

Why authors included the burnout only in the title despite measuring other conditions as anxiety, stress, depression, Resilience and others? Even should refer to it.

Also why authors involved medical and non-medical residents despite there were no difference between them regarding burnout syndrome at the end. (The overall prevalence of burnout in our sample was 33.3%, with no significant differences between medical residents and non-medical residents (35% vs. 31.6%, p = 0.191).

The Suggestive title (High prevalence of burnout and its associated factors among sample of Brazilians health residents during the COVID-19 pandemic).

Abstract:

Rewrite methods as: Healthcare residents were assessed for burnout (Oldenburg Burnout Inventory (OLBI)), resilience (brief resilient coping scale (BRCS), anxiety, stress and depression (depression, anxiety and stress scale (DASS-21) and Patient Health Questionnaire (PHQ-9)…………………etc.

Introduction:

• In early 2020, COVID-19, caused by a new coronavirus (SARS-CoV-2): write full name of SARS before writing the abbreviation (Severe acute respiratory syndrome (SARS).

• The objective of the present study was to evaluate the prevalence of burnout syndrome among health professionals in training, medical residencies (how to evaluate the prevalence, better to replace the action verb to determine) and write the goal if possible (what is the long-term outcome after determining the prevalence?

Material and methods:

• Mention the study design and study period (i.e. a descriptive online cross-sectional survey design was conducted from ……….to …….. (Write date beside month and year).

• Determine the type of sampling and the sampling technique (is it convenience sampling? How the authors reach to the healthcare residents all across the Brazil as mentioned in the abstract? Was it probability or non-probability sampling?

• How the sample size was calculated? Which software program was used (please write the reference) does they have list for all the public and private hospitals and health centers?

• Write in details about the study participants, selection criteria (inclusion and exclusion). Write the examples for medical and non-medical residents. Was there a pilot study?

• Concerning the questionnaire, was it structured as written? Or semi-structured?

What about the questions for the socio-demographic data and chronic diseases?

• In the data analysis and management: what about the normality test? The authors used parametric tests which mean that data was normally distributed. Also write the dependent and independent variables (please clarify and write in details)

Results:

• In table 1 the number of female (n=1025), male (n=285) isn’t equal to the total number (n= 1313 as mentioned in the table)

• Question (Increased risk for severe forms of COVID-19 in table 1) : what does this question mean, how the participant know whether they are at increased risk or not?

• Activity outside the residency program: give example for these activities as a note below the table.

• Where is the table for these results? (The mean (SD) age, for the overall sample, was 27.8 (4.4) years, and the mean scores for the instruments were as follows: OLBI-D, 2.8 (0.8); OLBI-E, 3.6 (0.7); OLBI Total, 3.2 (0.7); BRCS, 12.4 (3.8); DASS-21 depression, 15.3 (11.3); DASS-21 anxiety, 12.1 (10.3); DASS-21 stress, 20.3 (10.7); PHQ-9, 12.0 (6.5); perception of autonomy, 6.5 (2.1); and adequacy of the educational structure, 5.8 (2.5).

• Table 4: check the entire percentage % in the table. How it was calculated? for example

Number of male 94 (33%), number of female 342 (33.4%) how is it?

• Table 5: Differences Between Genders Regarding the Various Variables Studied. Why authors made this table despite it wasn’t from the objectives of the study to show the gender difference.

• Where the table for correlation as it is mentioned in the discussion and the abstract?

Discussion:

• The discussions missed data and interpretations on the regression analysis also the recommendations at the end.

Reviewer #2: #P 2 Line 37: after [18] There must be some word, it seems to be missing.

#P2 Line 46-53 : These statements should not be included in the introduction part with the discussion, theses should be moved to the discussion part. As well as line 64 onwards till the end of the introduction part.

Introduction part should be written in past tense as well the methodology part.

#P5 Research design should the written before the heading of "materials and methods"

Reviewer #3: The study appears to be well thought out and the manuscript is technically sound and well written. The sample size was quite large and the statistical analyses were appropriate and rigorously performed. All data underlying findings are available in the manuscript. The conclusions are also, supported by the data, and the findings are likely to be of global interest.

However, the authors should make a stronger justification for the paper and show how the paper contributes to new knowledge.

---

## [Author Response · Author response to Decision Letter 0]

17 Oct 2022

RESPONSE TO REWIEWERS:

1. Reviewer #1:

• Title: 

Questions 1 and 2:

1. Why authors included the burnout only in the title despite measuring other conditions as anxiety, stress, depression, Resilience and others? And

2. Why authors involved medical and non-medical residents despite there were no difference between them regarding burnout syndrome at the end? 

Answer: The absence of a significant difference between the medical and non-medical residents was a result, a finding of the research, considering that the work regimes and assignments are different in the different kinds off residents, due to the inherent differences in the residents professions. Initially, it was plausible to assume that there would be a difference between the groups, this was a question inherent to the reseach and after identifying this finding, it needed to be reported.

3. The Suggestive title (High prevalence of burnout and its associated factors among sample of Brazilians health residents during the COVID-19 pandemic).

Answer: We agree.

• Abstract: 

1. Rewrite methods as: Healthcare residents were assessed for burnout (Oldenburg Burnout Inventory (OLBI)), resilience (brief resilient coping scale (BRCS), anxiety, stress and depression (depression, anxiety and stress scale (DASS-21) and Patient Health Questionnaire (PHQ-9)…………………etc.

Answer: We agree with the suggested way, however, we would end up losing important information that would be suppressed with this new rewriting. This information would be: sample constitution - national; derivation of information: July to September 2020 and the mode of reach: digital forms that contained the assessment instruments. Removing that fragment would remove this elementary information, so we would like to keep it as it is written.

• Introduction: 

1. In early 2020, COVID-19, caused by a new coronavirus (SARS-CoV-2): write full name of SARS before writing the abbreviation (Severe acute respiratory syndrome (SARS).

Answer: Done as requested

2. The objective of the present study was to evaluate the prevalence of burnout syndrome among health professionals in training, medical residencies (how to evaluate the prevalence, better to replace the action verb to determine) and write the goal if possible (what is the long-term outcome after determining the prevalence?

Answer: For this research, the objective was to determine the prevalence of previously unknown burnout in this specific population and its associated factors. For the line of research that is still going on, there is a longitudinal research arm, which is not the target of this specific manuscrript. It has a long-term outcome that is to determine the incidence of burnout in individuals who did not have burnout at baseline and its risk factors. The practical implications of the prevalence findings are addressed in the discussion section.

• Materials and Methods: 

1. Mention the study design and study period (i.e. a descriptive online cross-sectional survey design was conducted from ……….to …….. (Write date beside month and year).

Answer: Everything the reviewer asked for has been entered. This is not just a descriptive study. This study uses an inferential approach (when we are looking for factors associated with burnout with an estimate of the probability of occurrence at random values) and makes inferences (in terms of the prevalence itself with confidence intervals). Although the analytical capacity of cross-sectional studies is admittedly lower than that of longitudinal studies, especially experimental studies, we effectively also used analytical procedures in this study, because groups were compared using statistical inferential analytical procedures with estimation of values and confidence intervals.

2. Determine the type of sampling and the sampling technique (is it convenience sampling? How the authors reach to the healthcare residents all across the Brazil as mentioned in the abstract? Was it probability or non-probability sampling?

Answer: The study adopted a convenience sampling and a non-probability sampling (the information is already included in the manuscript, in the paragraph that begins with the sentence: "The study adopted a convenience, non-probability (non-random) sampling approach"). The study adopted a convenience, non-probability (non-random) sampling approach. The way in which the residents were recruited was described in this same paragraph: “Recruitment occurred via e-mail, messages on social networks, posters in hospitals and the university hospital intranet containing QR codes with links to the survey forms. The codes and links were sent to the 7,215 residents of 40 university hospitals affiliated with the Brazilian Hospital Services Company (Empresa Brasileira de Serviços Hospitalares - EBSERH). EBSERH is a public company linked to the Ministry of Education established to manage federal university hospitals. Health professional residents at any healthcare institution in the country, considering the inherently unconstrained reach of the messages on social networks, were also allowed to participate”.

3. How the sample size was calculated? Which software program was used (please write the reference) does they have list for all the public and private hospitals and health centers?

Answer: The sample was calculated not exclusively aiming at the cross-sectional study, but at the size, the longitudinal line that is still in progress and that will be reported when finished. 

The G*Power 3.1.9.7 software was used. (already inserted in the references).

Initially, we had the e-mail of residents linked to EBSERH. As we also chose a mean of distributing the survey invitations that included social networks, considering that they do not have “barriers”, and also considering the inherently unconstrained nature of the reach of messages on social networks, we already anticipated the possibility that these invitations would reach residents outside the EBSERH conglomerate.

We didn't have a list of all the public and private hospitals in the country. We had a list of all active residents in the EBSERH network.

For this reason, the possibility of including participants from any institutions in the country that train human resources, in a postgraduate regime, in health área was expanded.

4. Write in details about the study participants, selection criteria (inclusion and exclusion). Write the examples for medical and non-medical residents. Was there a pilot study?

Answer: The eligibility criteria used for the inclusion of participants was: aged 18 years or above and postgraduate student in a medical residency or multidisciplinary residency program who has been designated for activities that involve direct patient care during the COVID-19 pandemic. The exclusion criteria have been defined as the explicit or assumed refusal to participate in the study as indicated by no response to telephone or electronic form interview attempts. The medical residents were physicians in training in several subspecialties of medicine such as: internal medicine, surgery, pediatrics, gynecology and obstetrics, orthopedics, cardiology, dermatology, among others. Non-medical residents were health professionals in training in other areas such as: nursing, nutrition, physical therapy, among others. There was a pilot study, a protocol entitle: Mental health and burnout syndrome among postgraduate students in medical and multidisciplinary residencies during the COVID-19 pandemic in Brazil: protocol for a prospective cohort study, that was published in 2021. And there will be a longitudinal follow up that is still in progress.

5. Concerning the questionnaire, was it structured as written? Or semi-structured?

Answer: The questionnaire was fully structured, as mentioned in the methodology.

6. What about the questions for the socio-demographic data and chronic diseases?

Answer: The most relevant chronic diseases were listed in the interview so that the participant could only select the option. There were also “OTHER” and “NONE” options. There were no open questions in the questionnaire.

7. In the data analysis and management: what about the normality test? The authors used parametric tests which mean that data was normally distributed. 

Answer: Parametric tests were used as a standard technique only to fulfill the formality in the evaluation of the statistical distribution of the variable. However, due to the large sample size and the central limit theorem, parametric tests that have more resources at their disposal could be perfectly applicable. For this reason, they were chosen. Formally, the normality tests indicated that the distributions were not normal, they were significant.

8. Also write the dependent and independent variables (please clarify and write in details)

Answer: The dependent variable was the burnout score measured by the OLBI instrument. For some analyses, this score was dichotomized between high and low, as described in the methodology section. The independent variables, the predictors of burnout, were gender, race, presence of illness, autonomy at work, perception of the pedagogical organization of the residency program, perception of adequacy of the availability of PPE, weekly workload, work activity outside the residency program, direct delivery of care to COVID 19 patients, and resilience.

Additionally, as secondary and exploratory objectives of the work, differences between the sexes and between the professional categories of the residents were tested regarding the same independent variables used for prediction and regarding the actual occurrence of burnout.

• Results: 

1. In table 1 the number of female (n=1025), male (n=285) isn’t equal to the total number (n= 1313 as mentioned in the table)

Answer: All calculated numbers and percentages indicated in the table refer to valid values, that is, those for which data were available for each of the variables. 

For example: men (n=285) and women (n=1025) and 13 individuals for whom there was no information for sex in the database, that is, missing data. And for each variable, there may be some degree of missing data.

2. Question (Increased risk for severe forms of COVID-19 in table 1) : what does this question mean, how the participant know whether they are at increased risk or not?

Answer: Participants who had chronic comorbidities (chronic heart disease, diabetes mellitus, chronic lung disease, chronic kidney disease, chronic arterial hypertension), immunosuppression, pregnancy, alcohol consumption, smoking, neoplasia in treatment and others, were classified as high-risk patients for severe forms of COVID-19.

3. Activity outside the residency program: give example for these activities as a note below the table.

Answer: In Brazil, medical residents are authorized to exercise professional activity in extra time to the residency program (NOTE: footnote inserted in table 1).

4. Where is the table for these results? (The mean (SD) age, for the overall sample, was 27.8 (4.4) years, and the mean scores for the instruments were as follows: OLBI-D, 2.8 (0.8); OLBI-E, 3.6 (0.7); OLBI Total, 3.2 (0.7); BRCS, 12.4 (3.8); DASS-21 depression, 15.3 (11.3); DASS-21 anxiety, 12.1 (10.3); DASS-21 stress, 20.3 (10.7); PHQ-9, 12.0 (6.5); perception of autonomy, 6.5 (2.1); and adequacy of the educational structure, 5.8 (2.5).

Answer: Initially, it was chosen to avoid the redundancy of information between text and table, as a general rule. It was identified that this portion of results could be accommodated in the text without needing to be reported in a new table.

5. Table 4: check the entire percentage % in the table. How it was calculated? for example number of male 94 (33%), number of female 342 (33.4%) how is it?

Answer: The denominators of the fraction that generated the results were inserted for better visualization of the table (inserted in table 4). Then a footnote was created clarifying how the percentages were calculated.

6. Table 5: Differences Between Genders Regarding the Various Variables Studied. Why authors made this table despite it wasn’t from the objectives of the study to show the gender difference.

Answer: Tables and analyzes were additionally performed, testing differences between genders and professional categories for the same variables used as predictors of burnout.

7. Where the table for correlation as it is mentioned in the discussion and the abstract?

Answer: Paragraph 6 of the discussion presents the findings in the logistic regression analysis, the independent predictors that persisted as relevant within the multivariate model.

• Discussion:

1. The discussions missed data and interpretations on the regression analysis also the recommendations at the end.

Answer: Paragraph 6 of the discussion presents the findings in the logistic regression analysis, the independent predictors that persisted as relevant within the multivariate model.

2. Reviewer #2:

• #P 2 Line 37: after [18] There must be some word, it seems to be missing.

There is no missing word in this sentence: "Almeida et al. [18] stated that females are more vulnerable to mental health problems, such as higher levels of stress, anxiety, depression and posttraumatic stress symptoms".

• #P2 Line 46-53 : These statements should not be included in the introduction part with the discussion, theses should be moved to the discussion part. As well as line 64 onwards till the end of the introduction part. Introduction part should be written in past tense as well the methodology part.

• #P5 Research design should the written before the heading of "materials and methods": It is already described in the methodology.

3. Reviewer #3:

• The authors should make a stronger justification for the paper and show how the paper contributes to new knowledge.

Although burnout in health professionals undergoing training is not new, it is possible that stressors related to the COVD-19 pandemic may contribute to the increased prevalence of this phenomenon in this population.

The high prevalence of burnout found showed that the reality is complex, and it is not possible to attribute the findings to a single factor. Through this study, it was possible to conclude that, in the context of the pandemic, the increase in working hours and inadequate physical and pedagogical structural conditions, in addition to the difficulty in adapting to a stressful situation, characterized by less resilience, contributed negatively to mental health of medical and multiprofessional residents.

The knowledge from this study, mainly related to the predictors associated with the development of burnout, may be useful for the elaboration of strategies to mitigate the damage caused by this phenomenon, propose actions that reduce the potential damages and the creation of better working conditions and health for this population, essential for the proper functioning of the establishments providing health services to the population.

---

## [Editor Report · Decision Letter 1]

19 Oct 2022

High prevalence of burnout syndrome among medical and nonmedical residents during the COVID-19 pandemic

PONE-D-22-10606R1

Dear Dr. Pinho,

We’re pleased to inform you that your manuscript has been judged scientifically suitable for publication and will be formally accepted for publication once it meets all outstanding technical requirements.

Kind regards,

Mohammad Hossein Ebrahimi

Academic Editor

PLOS ONE